# Predicted impact of banning nonessential, energy-dense food and beverages in schools in Mexico: A microsimulation study

Ana Basto-Abreu[1], Martha Carnalla[1], Francisco Reyes-Sánchez[1], Alan Reyes-García[1], Michelle M. Haby[2], Isabel Junquera-Badilla[1], Lianca Sartoris-Ayala[1], Juan A. Rivera[1], Barry M. Popkin[3], Tonatiuh Barrientos-Gutiérrez[1]*

**1** National Institute of Public Health, Center for Population Health Research, Mexico, **2** Division of Biological and Health Sciences, Department of Chemical and Biological Sciences, University of Sonora, Hermosillo, Mexico, **3** Department of Nutrition, Gillings School of Global Public Health, and the Carolina Population Center, The University of North Carolina at Chapel Hill, Chapel Hill, North Carolina, United States of America

* tbarrientos@insp.mx

**Data Availability Statement:** The data can be found in the following repository: https://zenodo.org/doi/10.5281/zenodo.8386075.

**Funding:** This work was funded by a Bloomberg Philanthropies grant (71206 to JAR; https://www.

## Abstract

### Background

Childhood obesity is a growing concern worldwide. School-based interventions have been proposed as effective means to improve nutritional knowledge and prevent obesity. In 2023, Mexico approved a reform to the General Education Law to strengthen the ban of sales and advertising of nonessential energy-dense food and beverages (NEDFBs) in schools and surroundings. We aimed to predict the expected one-year change in total caloric intake and obesity prevalence by introducing the ban of NEDFBs sales in schools, among school-aged children and adolescents (6 to 17 years old) in Mexico.

### Methods and findings

We used age-specific equations to predict baseline fat-free mass (FFM) and fat mass (FM) and then estimated total energy intake (TEI) per day. The TEI after the intervention was estimated under 4 scenarios: (1) using national data to inform the intervention effect; (2) varying law compliance; (3) using meta-analytic data to inform the intervention effect size on calories; and (4) using national data to inform the intervention effect by sex and socioeconomic status (SES). We used Hall's microsimulation model to estimate the potential impact on body weight and obesity prevalence of children and adolescents 1 year after implementing the intervention in Mexican schools. We found that children could reduce their daily energy intake by 33 kcal/day/person (uncertainty interval, UI, [25, 42] kcal/day/person), reducing on average 0.8 kg/person (UI [0.6, 1.0] kg/person) and 1.5 percentage points (pp) in obesity (UI [1.1, 1.9] pp) 1 year after implementing the law. We showed that compliance will be key to the success of this intervention: considering a 50% compliance the intervention effect could reduce 0.4 kg/person (UI [0.3, 0.5] kg/person). Our sensitivity analysis showed that the ban could reduce body weight by 1.3 kg/person (UI [0.8, 1.8] kg/person) and up to 5.4 kg/person (UI [3.4, 7.5] kg/person) in the best-case scenario. Study limitations include assuming that

bloomberg.org/). The funders had no role in study design, data collection and analysis, decision to publish, or preparation of the manuscript.

**Competing interests:** I have read the journal's policy and the authors of this manuscript have the following competing interests: Barry Popkin is a member of the Editorial Board of Plosmed.

**Abbreviations:** BAU, business-as-usual scenario; BMI, body mass index; FFM, fat-free mass; FM, fat mass; NEDFB, nonessential energy-dense food and beverage; pp, percentage points; SES, socioeconomic status; SSB, sugar-sweetened beverage; TEI, total energy intake; UI, uncertainty interval; WHO, World Health Organization.

obesity and the contribution of NEDFBs consumed at school remain constant over time, assuming full compliance, and not considering the potential effect of banning NEDFBs in stores near schools.

## Conclusions

Even in the most conservative scenario, banning sales of NEDFBs in schools is expected to significantly reduce obesity, but achieving high compliance will be key to its success.

## Why was this study done?

- School-based interventions have been recognized as effective means to improve nutritional knowledge and prevent obesity-related diseases.

- In December 2023, the Chamber of Representatives of Mexico approved an amendment that strengthens and updates the General Education Law (Article 75) and nutritional guidelines to ban the sales and advertising of nonessential energy-dense food and beverages (NEDFBs) in schools.

## What did the researchers do and find?

- We used age-specific equations to predict baseline fat-free mass (FFM) and fat mass (FM) and total energy intake (TEI) per day.

- We used microsimulation modeling to predict body weight and obesity prevalence of children and adolescents 1 year after implementing the intervention in Mexican schools.

- Our modeling study suggests that an important impact on obesity prevalence can be expected if the law is implemented and enforced as intended.

## What do these findings mean?

- If successful, this law could serve as an example beyond Mexico on how to achieve changes in body weight through school food regulation.

- An important limitation of our main scenario is that we assumed full compliance of schools with the law, yet lower compliance will reduce its impact. We also did not consider historical trends on obesity or NEDFBs consumed in schools during our 1 year simulation, and we considered only the ban impact inside schools, excluding effects near and outside schools.

## Introduction

Childhood obesity is a growing concern worldwide. In Mexico, the prevalence of obesity in children and adolescents is among the highest in the world, 18.6% and 18.2%, respectively [1]. In 2019, obesity was associated with 1.5% of all-cause mortality among Mexicans aged 10 to 24 years [2]. Children with overweight or obesity are more likely to become adults with obesity

[3] increasing their risk of developing chronic diseases such as type-2 diabetes, cardiovascular diseases, and cancer, significantly reducing life expectancy [4–7]. The healthcare costs of chronic diseases increase significantly during adulthood [8]; interventions to address childhood obesity are key for a healthy adulthood to reduce the disease and economic burden associated with obesity.

School-based interventions have been identified as effective policies to influence children's food consumption, nutritional knowledge, and obesity-related behaviors [9,10]. School-based interventions, such as nutritional standards for food and beverages sold in schools, have shown pooled reductions in unhealthy food intake of −0.17 servings/day and in sugar-sweetened beverages (SSBs) intake of −0.18 servings/day, resulting in reductions of 49 kcal/day in school and 79 kcal/day in total energy intake (TEI) [11]. Countries such as Australia, Bermuda, Chile, France, and Slovenia have implemented restrictions on the accessibility of junk food and sugary beverages within school facilities, including banning vending machines, implementing nutritional guidelines to improve school meals, and promoting water and milk as the only beverage options [12–16]. In Latin America, Chile implemented a package of obesity prevention initiatives that included front-of-package warning labels, child-directed food advertisement bans, and school regulations. Three years after the intervention, total sugars consumed by children at school decreased 11.8 percentage point (pp), saturated fats decreased 1.1 pp, and sodium decreased 10.3 mg/100 kcal. In adolescents, total sugars and saturated fats consumed at school decreased 5.3 pp and 1.5 pp, respectively [17]. In 2022, the Chilean Ministry of Education reported that school-age children decreased their prevalence of obesity by 4.8% relative to 2021 [18], which suggests promising results for countries implementing similar interventions, such as Mexico.

Mexico does not have a formal school meal program that covers all schools, except for a school breakfast program funded by the government that operates in less than 50% of elementary schools [19]. Instead, most schools have school cafeterias that are operated privately (the appointment and training of the provider is the responsibility of the Secretary of Education). In 2010, Mexico implemented national guidelines as a part of the Sectoral Health Program 2007–2012 [20] to regulate the preparation, sales, and distribution of foods and beverages that exceeded recommended nutritional standards in elementary and middle schools [21]. By 2014, through a reform of the General Education Law these guidelines became mandatory and sanctionable (Table A in S1 Appendix presents a list of prohibited food and beverages) [22]. The law provided an exception for Fridays, when all products were allowed. This law was poorly implemented: between 2012 and 2015, only 27% of food sold in schools met the nutritional criteria [23] and in 2018 in Northwest Mexico only 15% of schools had fully implemented the guidelines [24]. Low compliance has been explained by the poor dissemination and knowledge about the guidelines, lack of monitoring, enforcement and sanctions, failure to ban junk food advertising, and not applying restrictions to venues near schools [23,24]. Valuable lessons were learned from this experience: (1) increase parental involvement; (2) offer healthy substitutes for popular foods and beverages; (3) engage food delivery staff and teachers; (4) provide free drinking water; and (5) regulate advertising of unhealthy food in and around schools [23,24].

In December 2023, the Mexican Senate approved a reform to strengthen the General Education Law (Article 75) [25]. This reform law implied a structural change in the regulatory framework, emphasizing direct government involvement in assuring compliance. The approved legislation mandates the Secretary of Education to issue new General Guidelines for the preparation, sale, and distribution of prepared and processed foods and beverages in schools. Guidelines are under development, but the main changes to the law include the ban on advertising unhealthy food, expand the law to venues near schools, standardize the

nutritional criteria to front of package labels proposed in the General Law of Health (Art 212) (Table B in S1 Appendix) [26], promote healthier food alternatives prioritizing plain water and stronger sanctions for noncompliance. Also, to enhance compliance, the Secretary of Education will now monitor and ensure the enforcement of the new guidelines through school directors and supervisors, replacing the previous role of parental committees. With this changes, it is expected the law reform will promote better compliance. In the US, policies restricting SSB consumption in schools had higher impact when mandatory and when an administrative agency was designated as responsible for its implementation [27]. We aimed to estimate the expected impact of banning sales of nonessential energy-dense food and beverages (NEDFBs) in schools on total caloric intake and obesity prevalence of school-aged children and adolescents 1 year after its implementation, considering varying degrees of compliance.

## Methods

**Fig 1** summarizes each step of the modeling process. Briefly, we estimated baseline fat-free mass (FFM) and fat mass (FM) for the children and adolescents based on body mass index (BMI), age, and sex, using equations developed and validated by Deurenberg and colleagues [28]. This equation estimates the body fat percentage for children and adolescents, derived from a multiple linear regression. Baseline FFM and FM were used to predict the TEI per day depending on sex, age, and BMI category using a nonlinear function proposed by Hall and colleagues [29]. This TEI function considered children's growth and was derived from the sum of 2 components: energy balance (= energy intake–energy expenditure) and energy expenditure. We estimated the effect of the law on TEI in Mexican children under 4 different scenarios: (1) main scenario, using Mexican data to inform the intervention effect by education level; (2) sensitivity scenario varying law compliance; (3) sensitivity scenario, using meta-analytic data

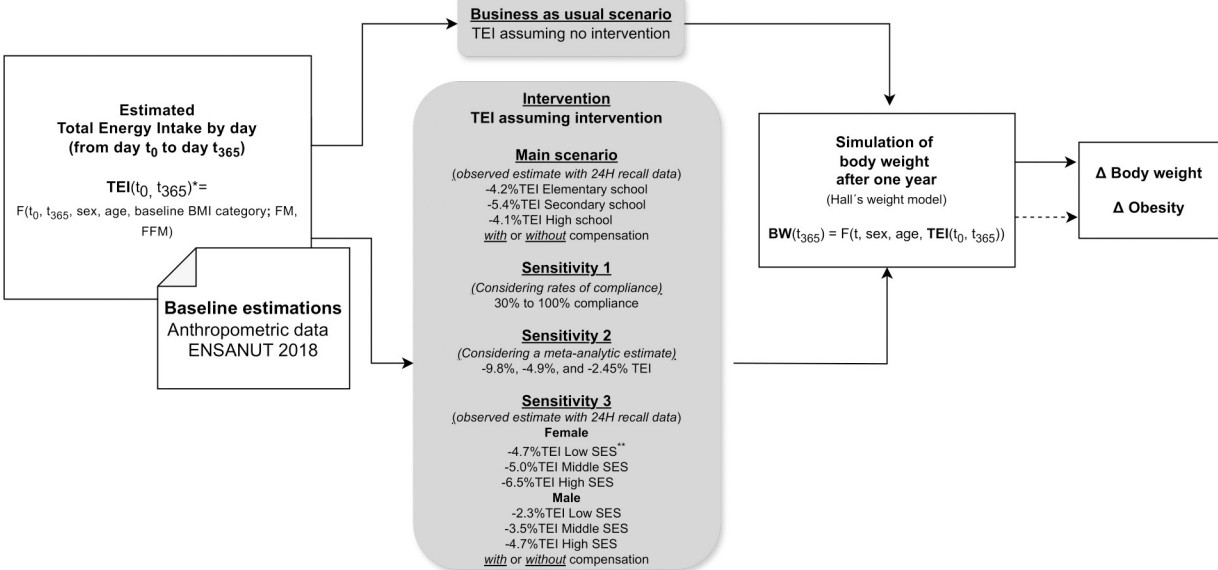

**Fig 1. Step-by-step modeling process, showing inputs and outputs of the model.** Changes in body weight and obesity prevalence are estimated by comparing the intervention scenario with BAU and 1 year after the intervention. *TEI $(t_0, t_{365})$ is a vector with the daily TEI from $t_0$ to $t_{365} = (t_0+365)$, considering sex, age, baseline (initial) BMI category, FM, and FFM. FM y FFM were estimated using the body fat percentage equation (%BF) developed and validated by Deurenberg and colleagues: FM = %BF*BW = (1.51*BMI– 0.7*Age– 3.6*Sex + 1.4)*BW/100 and FFM = BW–FM, where BW denote body weight and BMI represents body mass index. ** SES, socioeconomic status. BAU, business-as-usual scenario; FFM, fat-free mass; FM, fat mass; TEI, total energy intake.

to inform the effect of the intervention; and (4) using Mexican data to inform the intervention effect by sex and socioeconomic status (SES). Considering a business-as-usual scenario (BAU, no intervention) and the estimated intervention effect on TEI, we used Hall's microsimulation model to estimate the potential impact on body weight and obesity prevalence of children and adolescents that could be achieved 1 year after implementing the intervention in Mexican schools. A more detailed description of the modeling process is presented in the next subsections and in S1 Appendix. Model inputs are presented in Table C in S1 Appendix.

### Estimated total energy intake by day (TEI)

We estimated the daily TEI from day 0 ($t_0$) to day 365 ($t_{365}$) (one-year simulation period) for each individual using a nonlinear equation proposed by Hall and colleagues [29]. This equation accounts for daily increases in energy intake due to children's growth, dependent on sex, age, and baseline height and body weight. Hall's TEI equation is presented in S1 Appendix, section 3. We denote $TEI_i(t_0, t_{365})$ as the vector containing the estimated intakes from $t_0$ to $t_{365}$ for individual $i$ in the sample:

$$TEI_i(t_0, t_{365}) = F(t_0, t_{365}, sex_i, age_i, initial\ height_i\ and\ body\ weight_i) \tag{1}$$

The database needed for the baseline estimation was retrieved from the Mexican Health and Nutrition Survey (ENSANUT, from its Spanish acronym) 2018, specifically from anthropometric data for children and adolescents aged 6 to 17. ENSANUT is a cross-sectional, multistage, stratified, and cluster-sample survey representative at the national, regional, rural, and urban levels [30]. An overview of the baseline dataset with anthropometric sample sizes is described in section 2.1 in S1 Appendix. We calibrated Hall's TEI equation to our baseline sample. More details about the calibration process are presented in section 3 in S1 Appendix.

### Business as usual scenario (BAU)

We estimated changes in body weight and obesity after 1 year of simulation considering a BAU scenario, which assumed that children would continue to grow in the no-intervention scenario.

### Intervention scenarios

The BAU was compared to 3 intervention scenarios to estimate the expected impact of the law's implementation based on prior evidence.

**1. Main scenario using Mexican dietary data to inform the intervention effect.**   We used the 24-h dietary recall data from ENSANUT 2012 [31] to estimate the energy contribution of NEDFBs to TEI in schools, since this survey obtained information on place of consumption. NEDFBs were identified using the guidelines established in Mexico in 2020 for "excess" calories, added sugars, fats, trans fats, or sodium (Table G in S1 Appendix) [32]. According to ENSANUT 2012, NEDFBs intake in schools represents 4.2%, 5.4%, and 4.1% of the TEI in elementary, junior high, and high school students, respectively. Table C in S1 Appendix shows the contributions with confidence intervals. We simulated our main scenario by assuming 100% compliance, i.e., that these proportions of the TEI would be eliminated if the law effectively bans NEDFBs. We tested this scenario under 2 different assumptions:

a. *No energy compensation*, in which all energy consumed from NEDFBs in schools was subtracted from the TEI.

b. *Energy compensation*, in which a part of the reduced calories is compensated by the consumption of other food and beverages in schools. We estimated the expected percentage of

caloric compensation using the results from a previous study in Chile which estimated the consumption of regulated and nonregulated food and beverages at the household level after implementing a package of interventions, including warning labels, banning sales of unhealthy food and beverages at schools, and marketing restrictions [33]. After implementation, calories from regulated food and beverages decreased 33.1 and 16.3 kcal/capita/day, respectively, while calories from nonregulated items increased 26.8 in food and 6.3 kcal/capita/day in beverages. With these data, we estimated a caloric compensation of 81% for food (= 26.8/33.1) and 38.7% for beverages (= 6.3/16.3). The compensation rate considers compensation inside and outside schools, but this was applied to calories eaten inside schools and does not capture potential differences by age group.

**2. Sensitivity scenario varying compliance.** In the previous scenario, we assumed that all Mexican schools would fully implement the intervention. However, an evaluation of the implementation of the 2014 NEDFBs regulation guidelines in one Mexican State showed that only 15% of schools fully complied [24]. Due to the inclusion of enforcement and sanction mechanisms for the new law, we expect a higher compliance rate of at least twice the 2014 observed value (30%), with the possibility of reaching 100% depending on enforcement. For this analysis, we assumed that the compliance rate directly modifies weight reduction:

$$Weight\ reduction_i = i \times weight\ reduction_{main\ scenario}$$

Where i is the compliance rate with values = 30%, 40%, . . . or 100%

**3. Sensitivity scenario using meta-analytic data to inform the intervention effect.** To provide an alternative estimate the effect of the law, we conducted a meta-analysis based on published observational studies that implemented similar interventions, i.e., nutrition standards or banning NEDFBs in schools, compared to control. We calculated the mean difference in usual intake (kcal/day) using a random effects model and determined the percentage reduction in TEI. Section 4.3 in S1 Appendix details the included studies and the meta-analytic estimation. We estimated a 177 kcal reduction, representing a 9.8% reduction in TEI (95% CI [−13.4, −6.2] %) in children and adolescents after the intervention. As the included studies were observational (mostly before-after and cross-sectional studies), we expected an overestimation of the banning effect compared to experimental designs due to the lack of a suitable control group and potential confounding. Therefore, we considered 3 possible scenarios:

a. 100% of the effect (−9.8% TEI),

b. 50% of the effect (−4.9% TEI),

c. 25% of the effect (−2.45% TEI).

**4. Sensitivity scenario using Mexican dietary data to inform the intervention effect by socioeconomic status and sex.** For this scenario, the NEDFBs energy contribution was estimated by socioeconomic status (SES) and sex to assess the potential equity impact of the policy. As in the main scenario, we used the 24-h dietary recall data from ENSANUT 2012 to estimate the energy contribution of NEDFBs to TEI in schools and considered both the energy compensation and no energy compensation scenarios. For females, NEDFBs intake in schools represented 4.7%, 5.0%, and 6.5% of the TEI for the low, middle, and high SES, respectively. For males, those values were 2.3%, 3.5%, and 4.7% of the TEI for the low, middle, and high SES, respectively.

## Simulation of body weight after 1 year of intervention

Individual's body weight was simulated under both BAU and intervention scenarios using the Dynamics of Childhood Growth and Obesity Model (DCGO), proposed by Hall and colleagues [29]. Briefly, the DCGO is a nonlinear model that estimates changes in body weight over time (t) using a system of 2 differential equations to predict fat mass (FM(t)) and fat free mass (FFM(t)). To predict body weight (BW(t)), we considered the relation between energy intake rate and energy expenditure, adjusted by a growth term. All these variables consider children's characteristics (sex, age, initial height, and body weight) and other parameters that account for complex physiological processes during childhood and adolescence. The DCGO model was previously validated with experimental weight data [29] and calibrated to the Mexican population [34]. In this study, we re-calibrated and validated the model using Mexican data from the ENSANUT 2018. An overview of the microsimulation model and the calibration process is presented in sections 5 and 6 in S1 Appendix.

## Changes in body weight and obesity

Reductions in weight, BMI, and obesity prevalence were estimated by comparing each intervention scenario with the BAU scenario. To estimate BMI (in kg/m$^2$) after 1 year of intervention, we predicted each individual's height using the World Health Organization (WHO) growth reference (height for age) [35]. BMI was categorized in underweight, normal, overweight, and obesity based on the WHO's growth reference pattern for children and adolescents [36].

## Assumptions and other parameters

We assumed that all children and adolescents attend school and that the contribution of NEDFBs consumed at school remains constant over time. According to Hall's microsimulation model, physical activity progressively decreases with age [29]. For the BAU scenario, we assumed children's growth as the only factor for TEI daily changes without considering any previous trends or externalities that could cause a change in TEI. Additionally, we assumed that the height of children and adolescents would change according to WHO's height-for-age curves.

## Results

Table 1 presents the model results after 1 year of simulation for the BAU and main intervention scenarios. Considering compensation, children could reduce 33 kcal/person/day,

**Table 1. Estimates of caloric, body weight, BMI, and obesity changes 1 year after implementing the banning of NEDFBs in schools in Mexico.**

|  | BAU *[Reference] | Main scenario with energy compensation | | Main scenario without energy compensation | |
|---|---|---|---|---|---|
|  | Total (UI**) | Total (UI) | Change respect to BAU (UI) | Total (UI) | Change respect to BAU (UI) |
| *Energy intake (kcal/person/day)* | 2,316 [2,304, 2,328] | 2,283 [2,272, 2,295] | −33 [−42, −25] | 2,211 [2,200, 2,223] | −105 [−120, −90] |
| *Body weight (kg/person)* | 51.1 [50.6, 51.6] | 50.3 [50.1, 50.5] | −0.8 [−1.0, −0.6] | 48.6 [48.3, 49.0] | −2.5 [−2.9, −2.1] |
| *BMI (kg/m$^2$)* | 21.5 [21.3, 21.7] | 21.2 [21.1, 21.2] | −0.3 [−0.4, −0.2] | 20.4 [20.3, 20.6] | −1.1 [−1.2, −0.9] |
| *Obesity (pp)* | 16.1 [16.1, 16.1] | 14.6 [14.3, 15.0] | −1.5 [−1.9, −1.1] | 11.7 [11.1, 12.3] | −4.4 [−5.0, −3.9] |

*Business as usual (simulating 1 year under no intervention).

**UI = Uncertainty interval.

Scenarios are based on energy compensation and non-compensation for the consumption of banned foods inside schools.

BMI, body mass index; NEDFB, nonessential energy-dense food and beverage; pp, percentage points.

**Table 2. Body weight and obesity reduction 1 year after the intervention varying compliance.**

| | | Body weight (kg/person) | | Obesity (pp) | |
|---|---|---|---|---|---|
| | | Considering energy compensation | Not considering energy compensation | Considering energy compensation | Not considering energy compensation |
| Compliance rate (%) | 100 | −0.8 [−1.0, −0.6] | −2.5 [−2.9, −2.1] | −1.5 [−1.9, −1.1] | −4.4 [−5.0, −3.9] |
| | 90 | −0.7 [−0.9, −0.5] | −2.2 [−2.6, −1.9] | −1.2 [−1.7, −1.0] | −4.1 [−4.5, −3.5] |
| | 80 | −0.6 [−0.8, −0.5] | −2.0 [−2.3, −1.7] | −1.1 [−1.5, −0.9] | −3.5 [−4.0, −3.1] |
| | 70 | −0.6 [−0.7, −0.4] | −1.7 [−2.0, −1.5] | −1.0 [−1.3, −0.8] | −2.9 [−3.5, −2.7] |
| | 60 | −0.5 [−0.6, −0.4] | −1.5 [−1.7, −1.3] | −0.9 [−1.1, −0.7] | −2.6 [−3.0, −2.3] |
| | 50 | −0.4 [−0.5, −0.3] | −1.2 [−1.4. −1.0] | −0.8 [−1.0, −0.5] | −2.2 [−2.5, −1.9] |
| | 40 | −0.3 [−0.4, −0.2] | −1.0 [−1.2, −0.8] | −0.6 [−0.8, −0.4] | −1.8 [−2.0, −1.6] |
| | 30 | −0.2 [−0.3, −0.2] | −0.7 [−1.0, −0.6] | −0.5 [−0.6, −0.3] | −1.4 [−1.5, −1.2] |

Results are presented for body weight (kg) and obesity (percentual points) considering compliance rates from 30% to 100%.

pp, percentage points.

meaning a 0.8 kg reduction in weight (uncertainty interval, UI, [0.6, 1.0] kg), 0.3 kg/m² in BMI (UI [0.2, 0.4] kg/m²), and 1.5 pp in obesity prevalence (UI [1.1, 1.9] pp). That change in obesity prevalence represents a relative reduction of 9.3% compared to BAU (= 1.5*100/16.1). Without energy compensation, children could reduce 105 kcal/person/day (UI [90, 120] kcal/person/day), which translates into a 2.5 kg reduction in weight (UI [2.1, 2.9] kg), 1.1 kg/m² in BMI (UI [0.9, 1.2] kg/m²), and 4.4 pp (UI [3.9, 5.0] pp) in obesity prevalence, representing a 27.3% reduction compared to BAU (= 4.4 *100/16.1).

Table 2 presents the results of the sensitivity analysis varying the compliance rate of schools. It shows the impact on the changes of 2 outputs of interest: body weight and obesity. We found that, by varying compliance rates between 30% and 100%, weight reductions could range from 0.2 to 0.8 kg with compensation and from 0.7 to 2.5 kg without compensation. Obesity reductions could range from 0.5 to 1.5 pp with compensation and from 1.4 pp to 4.4 pp without compensation.

Fig 2 shows the results of the sensitivity analysis using meta-analytic estimates of the reduction in energy intake in Mexican school children compared to the main scenario. It shows the impact of the law banning NEDFBs in schools on the change in 2 outputs of interest: body weight and obesity prevalence. For weight, the sensitivity analysis showed reductions ranging from 1.3 to 5.4 kg in body weight and from 2.3 to 9.3 pp in obesity prevalence. Using the 100% effect of the meta-analysis, we found significantly larger reductions in body weight and obesity than in both main scenarios, with and without compensation. When using 75% and 50% of the effect of the meta-analysis, the results were not different from the main scenario without compensation.

Fig 3 shows results stratified by sex and SES. We found that the expected impact is estimated to be higher in females, mainly in higher SES, with an expected reduction of 1.1 kg/person considering compensation, which translates into 2.1 pp obesity reduction. Among males, we found a higher expected impact among high SES than low SES with an expected reduction of 0.9 kg/person considering compensation, which translates into 1.8 pp obesity reduction.

## Discussion

We aimed to estimate the expected impact of banning NEDFBs inside schools 1 year after implementation. Considering intake compensation, children could reduce their daily energy consumption by approximately 33 kcal/day/person, which translates into an average reduction

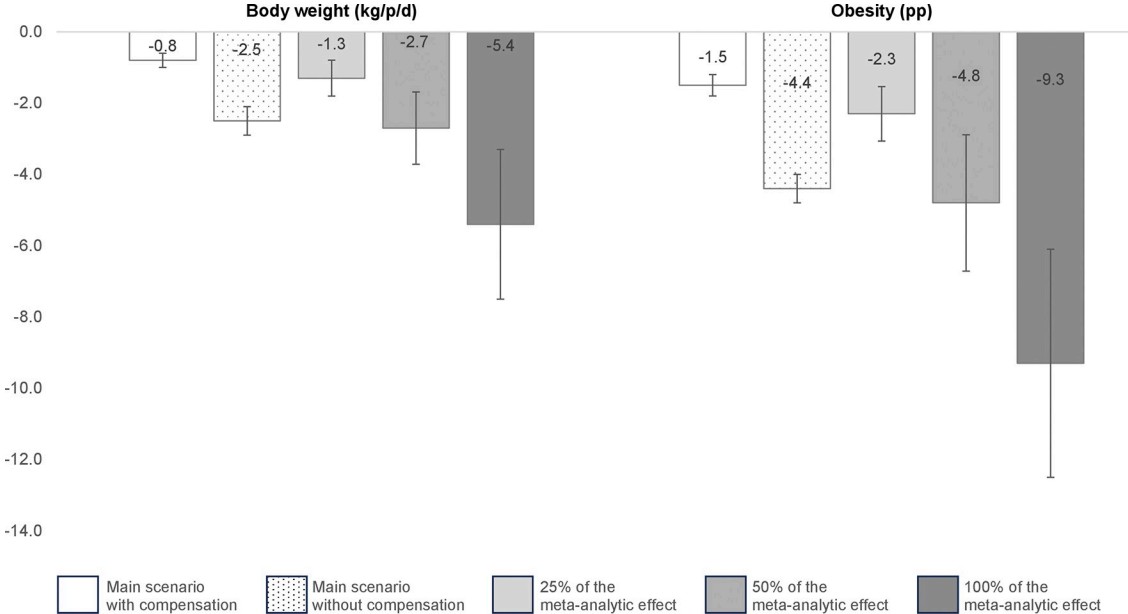

**Fig 2. Uncertainty captured by the sensitivity analysis using the meta-analytic estimated reduction of the impact of banning NEDFBs in schools.** The meta-analysis considered observational studies evaluating interventions on nutrition standards and the banning of NEDFBs in schools. We considered 3 possible scenarios: 100% of the effect (−9.8% TEI), 50% (4.9% TEI), and 25% (2.45% TEI) considering that observational data could overestimate the total effect. The main scenarios with and without energy compensation corresponded to average reductions of 1.4% and 4.5% of TEI, respectively. NEDFB, nonessential energy-dense food and beverage; pp, percentage point; TEI, total energy intake.

of 0.8 kg in weight and 1.5 pp in obesity prevalence. The estimated impact could be higher if the effect size of the intervention is equal to the meta-analytic results using observational data but could be lower if not all schools comply with the intervention. Even in the most conservative scenario, we expect the intervention to reduce obesity, but achieving high compliance will be key to its success.

The expected impact on BMI of banning NEDFBs in schools in our analysis is higher compared to prior studies evaluating similar interventions using observational data. The most comparable study was conducted in children aged 12 to 15 years in Canadian schools, where junk food was banned. Five to 8 years after the intervention, they observed a mean reduction in BMI of 0.32 kg/m$^2$ (95% CI [0.03, 0.62]) [37]. That change would represent an average yearly change of −0.06 kg/m$^2$, although weight change is nonlinear, and a larger decrease is to be expected in the first year. While temporalities are difficult to align, this is lower than our predicted 1-year BMI change of −0.3 kg/m$^2$ (UI [−0.4, −0.2] kg/m$^2$) for children aged 6 to 17 years. Baseline caloric consumption and the proportion of calories consumed in school are likely the main reason for this difference, although caloric data was not provided in the Canadian report. A Brazilian study showed reductions in body composition among girls aged 11 to 15 years, although the results are expressed in BMI z-scores for age [38], hindering comparisons. However, the direction of the effect is the same in response to regulations on the sale of processed food and beverages in schools. Both observational studies and our simulation results showed that banning NEDFBs in schools should lead to a reduction in BMI, although the magnitude of the effect relies on baseline caloric consumption, different settings and different assumptions in our simulation model (compliance rate and compensation rate).

Our prediction of −33 kcal/person/day (UI [−42, −25] kcal/person/day) from banning NEDFBs, which translates into a −0.8 kg (UI [−1.0, −0.6] kg), is higher than the expected

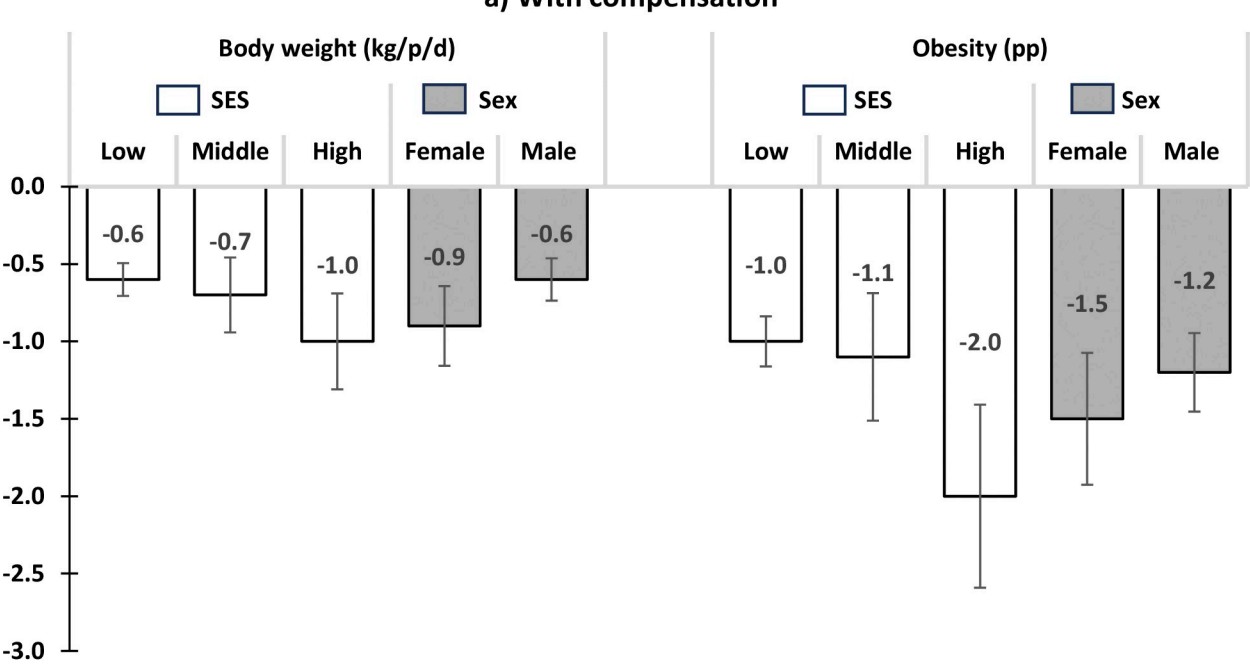

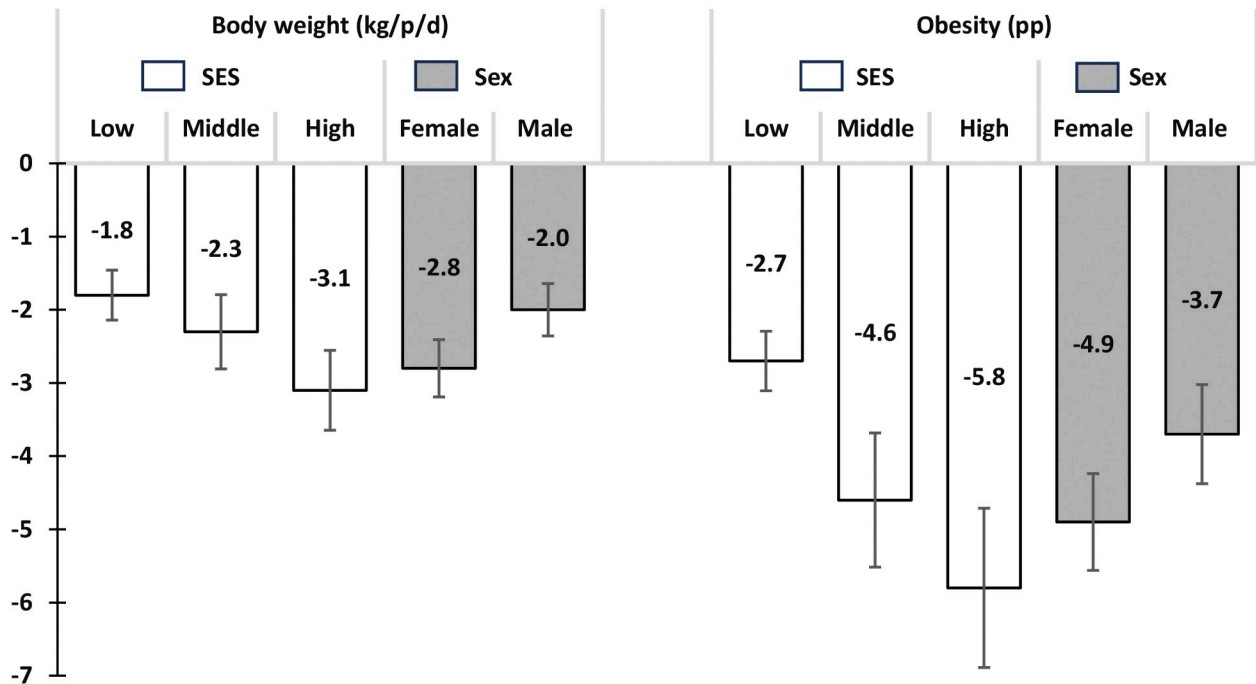

**Fig 3. Predicted impact in body weight, and obesity by SES and sex 1 year after implementing the banning of NEDFBs in schools in Mexico.**
NEDFB, nonessential energy-dense food and beverage; SES, socioeconomic status.

impact of the 10% SSBs tax and 8% junk food tax on Mexican children, but lower than both tax interventions combined. Taxes on SSBs and junk food were expected to reduce 17.6 kcal/person (UI [16.3, 18.8] kcal/person) and 17.4 kcal/person (UI [16.4, 18.4] kcal/person),

respectively, 1 year after their implementation [34,39]. This translates into body weight reductions of 0.42 kg (UI [0.39, 0.45] kg) for the SSBs tax and 0.40 kg (UI not available) for the junk food tax. We used a similar microsimulation model, a similar age group, and targeted the same food as both previous studies, but with a different intervention (tax versus school ban). In all 3 cases, weight reduction is simulated for 1 year, although the nonlinear equation used in all studies suggests that further impacts on body weight after the first year of simulation are to be expected [40]. Empirical evidence from an online survey in adolescents aged 13 to 17 years in Mexico showed a reduction of 38.7% in the purchase of NEDFBs after the implementation of the front-of-package labeling [41]. However, the relative reduction does not allow us to quantify the reduction in kcal to properly compare the results with our study.

Some limitations must be acknowledged. First, the total impact of the intervention may be underestimated because we are using 24-h nutritional recall data from 2012 to estimate the calories from junk food consumed at school, which is known to underestimate unhealthy food consumption. Also, for children under 12 years of age, 24-h nutritional recall data obtains information from parents or guardians, who may not be well informed about their child's consumption patterns at school. Still, ENSANUT is representative at the national level, allowing us to make population estimates generalizable to the Mexican context. Results may be overestimated because we are assuming that all children and adolescents go to school, but in the 2019 to 2020 cycle, 98.7% of 6- to 12-year-olds, 95.9% of 13- to 15-year-olds, and 75.5% of 16- to 18-year-olds were enrolled in school [42]. Given the uncertainty of the compliance rate, we tested different compliance levels, but we assumed that it would be at least twice the compliance of the 2014 guidelines implemented in Mexico (15%) [24]. We are assuming that the banning effect and compliance rate are equal for public or private schools. Evidence exists that private schools had better compliance with the regulation than public schools [24], but the overall effect is likely to be small, as most Mexican children attend public schools [42]. Overestimation could also arise from energy compensation, either from allowed food and beverages in school or through NEDFBs consumed outside of schools. We considered substantial levels of food compensation (81%) and beverages (39%), based on household-level estimates after the Chilean policies on food labeling, marketing, and sales in schools [33]. Another study from Chile after implementing the same policies did not report caloric compensation, but compensation for specific nutrients. They reported an 80% sugar compensation for children, but complete sugar compensation for adolescents [17]. Our model considers a growth-related increase in TEI (Table I in S1 Appendix) but does not include an increase in TEI due to past obesity trends over time. This assumption should have little effect on estimates, given the one-year span of our simulation. Finally, we are not considering the potential effect of banning junk food near schools, as our study only included the expected impact inside schools.

The school context is very important for implementing and impacting health interventions. Our sensitivity analysis using a meta-analytical input, showed a higher expected effect of the intervention. However, studies included were observational, often lacked a control group and were subject to residual confounding. Also, studies in the meta-analysis were conducted in high-income countries, where infrastructure and organization of school food services is better and where compliance could also be higher than in Mexico. Compliance will play a key role in the success of any law targeting the school food environment. Previous experience with the guidelines issued in 2010 showed that the lack of dissemination and training, the lack of regulations outside and near schools, and the lack of monitoring were key elements for low compliance [23,24]. Furthermore, school cafeterias share a small percentage of profits with schools, which are used to buy essential supplies, improve facilities, or finance school activities; thus, food sales are an economic incentive for schools [24]. With the 2023 modification to the law, some of these limitations are addressed, including the promotion of endemic and natural

foods from each region to reduce the incentive to sell processed foods, as well as a mechanism for implementation, evaluation, monitoring, and sanctions for noncompliance. However, considering that the degree of compliance is still uncertain and may vary for different reasons, including possible economic incentives for schools to sell processed foods, we evaluated its impact in our sensitivity analysis, showing that if all schools comply with the law, the intervention is predicted to reduce 0.8 kg/person, but if only 30% of schools comply with it, the intervention is predicted to reduce 0.2 kg/person.

Our modeling is based on the best available evidence of the potential impact of the new law. It suggests that an important impact on obesity prevalence can be expected if the law is implemented and enforced as intended, but that lower effects will be expected if enforcement is suboptimal. Our study highlights the need for better evidence about the actual impact of government laws and policies such as these to determine their effectiveness and whether adjustments are needed. If the predicted impact of the new law is achieved it could serve as an example beyond Mexico of how to achieve changes in childhood obesity.

Policy actions based on improving the nutritional quality of foods available in schools have been recognized as key strategies for improving children's diet [43,44]. The estimated obesity reduction ranged from 1.0 to 3.3 pp on average 1 year after the law's implementation in our main scenario. Establishing laws to limit the consumption of NEDFBs in schools promotes healthy food choices and the early adoption of healthy eating habits. This will allow students to develop a lifelong appreciation for nutritious foods, improve health outcomes, and reduce the risk of chronic diseases (such as type-2 diabetes, hypertension, hyperlipidemia, and depression) [45–47] that have major economic implications for health systems and society. This law could be part of a long-term strategy to prevent and treat childhood overweight and obesity, leading to healthy, productive adults and reducing the predicted healthcare burden worldwide. By creating a healthier school environment, we can help shape the future health and well-being of our children and communities.

## Supporting information

**S1 Appendix. Additional information on the simulation model and assumptions.** (PDF)

## Acknowledgments

We would like to thank to Gabriela García Chávez for her valuable insights on this manuscript.

## Author Contributions

**Conceptualization:** Ana Basto-Abreu, Alan Reyes-García, Isabel Junquera-Badilla, Juan A. Rivera, Barry M. Popkin, Tonatiuh Barrientos-Gutiérrez.

**Data curation:** Martha Carnalla, Francisco Reyes-Sánchez, Alan Reyes-García, Michelle M. Haby.

**Formal analysis:** Ana Basto-Abreu, Martha Carnalla, Alan Reyes-García, Tonatiuh Barrientos-Gutiérrez.

**Funding acquisition:** Juan A. Rivera, Tonatiuh Barrientos-Gutiérrez.

**Investigation:** Ana Basto-Abreu, Francisco Reyes-Sánchez, Michelle M. Haby, Isabel Junquera-Badilla, Juan A. Rivera.

**Methodology:** Ana Basto-Abreu, Martha Carnalla, Francisco Reyes-Sánchez, Alan Reyes-García, Isabel Junquera-Badilla, Juan A. Rivera, Barry M. Popkin, Tonatiuh Barrientos-Gutiérrez.

**Project administration:** Lianca Sartoris-Ayala, Tonatiuh Barrientos-Gutiérrez.

**Resources:** Tonatiuh Barrientos-Gutiérrez.

**Supervision:** Ana Basto-Abreu, Martha Carnalla, Michelle M. Haby, Tonatiuh Barrientos-Gutiérrez.

**Validation:** Juan A. Rivera, Tonatiuh Barrientos-Gutiérrez.

**Visualization:** Lianca Sartoris-Ayala, Barry M. Popkin, Tonatiuh Barrientos-Gutiérrez.

**Writing – original draft:** Ana Basto-Abreu, Martha Carnalla, Francisco Reyes-Sánchez, Alan Reyes-García, Lianca Sartoris-Ayala, Juan A. Rivera, Tonatiuh Barrientos-Gutiérrez.

**Writing – review & editing:** Ana Basto-Abreu, Martha Carnalla, Francisco Reyes-Sánchez, Alan Reyes-García, Michelle M. Haby, Isabel Junquera-Badilla, Lianca Sartoris-Ayala, Juan A. Rivera, Barry M. Popkin, Tonatiuh Barrientos-Gutiérrez.

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
