## [Editor Report · Decision Letter 0]

14 Nov 2023

Dear Dr Barrientos-Gutierrez, 

Thank you for submitting your manuscript entitled "Expected impact of banning non-essential, energy-dense food and beverages in schools in Mexico: a microsimulation study" for consideration by PLOS Medicine.

Your manuscript has now been evaluated by the PLOS Medicine editorial staff as well as by an academic editor with relevant expertise and I am writing to let you know that we would like to send your submission out for external peer review.

Please re-submit your manuscript within two working days, i.e. by Nov 16 2023 11:59PM.

Kind regards,

Katrien G. Janin, PhD

Senior Editor

PLOS Medicine

---

## [Decision Letter · Decision Letter 1]

16 Dec 2023

Dear Dr. Barrientos-Gutierrez,

Thank you very much for submitting your manuscript "Expected impact of banning non-essential, energy-dense food and beverages in schools in Mexico: a microsimulation study" (PMEDICINE-D-23-03317R1) for consideration at PLOS Medicine. 

[LINK]

In light of these reviews, I am afraid that we will not be able to accept the manuscript for publication in the journal in its current form, but like to consider a revised version that addresses the reviewers' and editors' comments. We cannot make any decision about publication until we have seen the revised manuscript and your response, and we plan to seek re-review by one or more of the reviewers. 

We expect to receive your revised manuscript by Jan 08 2024 11:59PM. Please email us (plosmedicine@plos.org) if you have any questions or concerns.

We look forward to receiving your revised manuscript. 

Sincerely,

Katrien Janin, PhD

PLOS Medicine

plosmedicine.org

Comments from the Academic Editor:

For the meta-analysis, I'm surprised by an intervention that reduces total energy intake by 10% - this is substantial and I think unrealistic. But I think that the authors do a good job of not over-emphasising this finding from sensitivity analysis. It would be nice to have some discussion of how realistic the estimation from the meta-analysis really is. It just seems unlikely to me that children would reduce energy intake by 10%long term.

The paper is poorly contextualised. I'd like to see more discussion of: limitations of meta-analysis as above, why compliance with the existing regulations is low and why it might be expected to improve with the new regulations (compliance seems the key to the whole paper - both in terms of the justification for the intervention and results of sensitivity analysis), how the impacts of this intervention compare to other potential approaches to obesity (there is a little bit of this, but I'd suggest expanding further). Related, I'd also like to see the UI in table 2 - it's not clear at what point the effect is indistinguishable from null.

Comments from the Editorial Team:

We would like to ask the authors the following:

i) In line with the academic editor and reviewer comments, please improve the contextualisation of the presented research

ii) for the ‘Sensitivity scenario using meta-analytic data to inform the intervention effect’ in the method section please detail your search criteria and search period, inclusion and exclusion criteria and the basis upon which you selected studies to be included in the meta analysis. We note that studies were mainly sourced from ‘Effectiveness of school food environment policies on children’s dietary behaviors: A systematic review and metaanalysis.’ publication. We also note you have added one study, but how was this study sourced? Did you perform a systematic literature review to update the Micha study? This is unclear to us. Please provide more details (beyond those details in Section 3.3 in S1 Appendix) and ask you to move pertinent information inot the main manuscript. We also like more details on how you arrived at the estimated 177-kcal reduction (9.8% reduction in TEI), especially given that this substantially differs from the data derived from ENAUD. In the discussion, please add more details on why these estimates differ so much, and how realistic this scenario (or not) may be. Once again, we appreciate you have added that this is likely an overestimate, but ask a bit more detail and ask why you think this model - if likely an overestimate - adds to the discussion).

iii) In the Discussion, be more explicit about how the effect sizes compare to prior studies; and contextualise findings including with regard to compensatory calorie intake and weight regain.

iv) In the Discussion, also clearly point out what this adds to prior knowledge and say something about what the findings are useful for (and what they are not useful for), and expand on what these findings mean beyond Mexico. 

GENERAL: 

Please provide 95% CIs and p values for all results where appropriate (including the abstract), check and amend throughout. We suggest reporting statistical information in the following format: ‘x’; (95% CI [‘y’,’ z’] p value) and use commas as opposed to hyphens (as these can be confused with negative values) to separate upper and lower bounds. For p values, please report as p<0.001 and where higher as 'p=0.002'. Please add the statistical method used to your method section. We also invite you to report p values to consistently to the third decimal digit - thousandths.

STUDY DESIGN: MODELLING STUDIES 

Of all authors who submit a modelling study we ask for inclusion of specific items, derived from Geoffrey P Garnett, Simon Cousens, Timothy B Hallett, Richard Steketee, Neff Walker. Mathematical models in the evaluation of health programmes. (2011) Lancet DOI:10.1016/S0140-6736(10)61505-X.

Please ensure all the items listed below are included with your manuscript. Please review the list below and confirm/revise as necessary:

i) Please provide a diagram that shows the model structure, including how the disease natural history is represented, the process and determinants of disease acquisition, and how the putative intervention could affect the system.

ii) Please provide a complete list of model parameters, including clear and precise descriptions of each parameter, together with the values or ranges for each, with justification or the primary source cited, and important caveats about the use of these values noted.

iii) Please provide a clear statement about how the model was fitted to the data [including goodness-of-fit measure, the numerical algorithm used, which parameter varied, constraints imposed on parameter values, and starting conditions].

iv) For uncertainty analyses, please state the sources of uncertainties quantified and not quantified [can include parameter, data, and model structure].

v) Please provide sensitivity analyses to identify which parameter values are most important in the model. Uncertainty estimates seek to derive a range of credible results on the basis of an exploration of the range of reasonable parameter values. The choice of method should be presented and justified.

vi) Please discuss the scientific rationale for this choice of model structure and identify points where this choice could influence conclusions drawn. Please also describe the strength of the scientific basis underlying the key model assumptions.

TITLE: We suggest you replace ‘expected’ by ‘predicted’ (Predicted impact of banning non-essential, energy-dense food and beverages in schools in Mexico: a microsimulation study). 

AUTHORS SUMMARY: At this stage, we ask that you include a short, non-technical Author Summary of your research to make findings accessible to a wide audience that includes both scientists and non-scientists. The Author Summary should immediately follow the Abstract in your revised manuscript. This text is subject to editorial change and should be distinct from the scientific abstract. Please see our author guidelines for more information: https://journals.plos.org/plosmedicine/s/revising-your-manuscript#loc-author-summary

Ideally each sub-heading should contain 2-3 single sentence, concise bullet points containing the most salient points from your study.

In the final bullet point of ‘What Do These Findings Mean?’ Please include the main limitations of the study in non-technical language.

DISCUSSION:

Please present and organize the Discussion as follows: a short, clear summary of the article's findings; what the study adds to existing research and where and why the results may differ from previous research; strengths and limitations of the study; implications and next steps for research, clinical practice and/or public policy implications; followed by a one-paragraph conclusion. Please remove all subheadings within your Discussions

Comments from the reviewers:

Reviewer #1: The manuscript "Expected impact of banning non-essential, energy-dense food and beverages in schools in Mexico: a microsimulation study" describes the expected effect of a new law (which is currently in the legislative process) banning the sale and advertising of unhealthy foods inside schools and surrounding, in Mexico. Using previously proposed equations for estimating total daily energy intake, and nationally representative data on body weight, weight status and intake of unhealthy foods/beverages at schools, authors modeled the effect of the policy on energy intake, body weight, body mass index and prevalence of obesity among school-age children, considering different scenarios (i.e., dietary compensation, school compliance, effect size of the decrease in energy intake). Simulating the effect of a policy which is under legislative discussion is a good strategy to inform policy-makers' decision, and a good example of the potential use of scientific investigation when it is well conducted, in a relevant topic, and timely done/published. Moreover, policies aiming at improving school food environments are a especially relevant strategy to address the obesity pandemic. Therefore, this manuscript is highly relevant. 

The manuscript is well written, the methodology is clearly explained and the details are presented largely in the supplementary file; results are clear and relevant. The document reflects a well conducted analysis. From my perspective, authors fail in presenting the current situation of school food environments in the introduction, and contrasting it with the new scenario (the one expected after the policy implementation). I believe one difference between the current and the expected situation is that in the former nutritional standard are optional, whereas the latter implies a mandatory situation (i.e., a ban). Additionally, currently the standards apply to preparation, sale and distribution of foods, whereas the new scenario will apply just to food sales and marketing (is there a food feeding program in Mexico?) Finally, I believe the nutritional criteria for identifying unhealthy foods are different between both scenarios, given -I understood- the new regulation uses the same cutoffs used for defining foods that should have a warning label, which were developed in the last few years. All these aspects should be clearly stated in the introduction (of course, I might be wrong in my interpretation). 

Specific comments

The quality of Figure 1 should be improved.

"BAU" should be presented earlier in the text, the first time wen "business as usual" is mentioned.

Data from ENSANNUT 2012 were used for estimating the dietary share of unhealthy foods eaten at schools, whereas ENSANNUT 2018 was used for obtaining the weight status data and estimating total energy intake; please provide the rationale for doing so.

Authors estimated a compensation scenario based on compensation data from household food purchases obtained after implementing the Chilean food labelling regulation. A different article evaluating the Chilean regulation showed a greater compensatory effect (for energy from total sugars and saturated fats) for the dietary improvement obtained at schools among adolescents (dietary compensation made at home and other locations): https://ijbnpa.biomedcentral.com/articles/10.1186/s12966-023-01445-x. Given total energy intake is not reported in the article, data cannot be used for the current analyses; however, authors could discuss these data, acknowledging a potential underestimation of the dietary compensation. 

Legend could be improved for Figure 2 (the squares are too little to see the colors). 

The second paragraph of discussion reads: "The Canadian model assumes the intervention increases linearly over time; our model assumes a non-linear function of calories on body mass over time, and a significant proportion (around 50%) of the intervention effect will be achieved during the first year and then decrease progressively." I am confused with this statement. I assume "our model" refers to the one used in the current analysis; in that case, it is relevant the non-linear simulation model is presented earlier in the text and in a clearer way. 

In the third paragraph, authors commented on modeling analyses performed previously for Mexican taxes on foods and sugary-sweetened beverages. It would be relevant to discuss whether these estimations were confirmed with real

---

## [Decision Letter · Decision Letter 2]

19 Mar 2024

Dear Dr. Barrientos-Gutierrez,

Thank you very much for re-submitting your manuscript "Predicted impact of banning non-essential, energy-dense food and beverages in schools in Mexico: a microsimulation study" (PMEDICINE-D-23-03317R2) for review by PLOS Medicine.

I have discussed the paper with my colleagues and the academic editor and it was also seen again by the reviewers. I am pleased to say that provided the remaining editorial and production issues are dealt with we are planning to accept the paper for publication in the journal. 

Given the minor nature of the remaining comments (see comment from reviewer #3), we expect to receive your revised manuscript within the next few days week (Friday, 22nd of March, 2024). Please email me (kjanin@plos.org) if you have any questions or concerns.

[LINK]

If you have any questions in the meantime, please contact me (kjanin@plos.org) or the journal staff on plosmedicine@plos.org.  

We look forward to receiving the revised manuscript by Mar 22 2024 11:59PM.   

Sincerely,

Katrien Janin, PhD

Senior Editor 

PLOS Medicine

plosmedicine.org

Note from the editors:

We like to thank the authors for addressing fully our comments, and for the the additional clarity you have provided provided in the manuscript. (but please do see the outstanding comment from reviewer #3)

As a general comment, please note that supplementary materials are not checked and will be posted as supplied by the authors. Therefore, please double check and ensure you appropriate cite the Supplementary Information in your manuscript. Please cite your Supporting Information as outlined here: https://journals.plos.org/plosmedicine/s/supporting-information - Please note you may use almost any description as the item name of your supporting information as long as it contains an "S" and number. For example, “S1 Appendix” and “S2 Appendix,” “S1 Table” and “S2 Table.

I look forward to receiving the revised version of your manuscript, and issuing the editorial accept for you paper, 

Best wishes,

Katrien Janin

(kjanin@plos.org)

Comments from Reviewers:

Reviewer #2: In my opinion, the authors have adequately addressed my and other reviewers' comments and concerns. Congratulations!

Reviewer #3: Dear authors,

Thank you for taking the time to address my comments. Your responses are well considered and I thank you for the additional clarity provided in the manuscript to address these points. 

I would like to ask for one further clarification please, regarding "A brief overview of the school food system in Mexico to provide a picture of any potential commercial influences that might influence compliance with the new restrictions on sales and advertising of unhealthy food inside schools and surroundings e.g. is school food prepared by in-house caterers, centrally provided (e.g. local or national catering provider) or by private companies?" I am still not clear on the school food system in Mexico and potential additional motives for low compliance - for example, is the food sold in schools done so through private contracts with catering companies (in which case there may be a strong profit motive for non-compliance via increased sales) or is food sold via local/national/school catering services (i.e. not-for-profit)? This may help to provide some context around Mexico's school food system for those unfamiliar with it.

Unfortunately I am still finding the figure 1 to be of low resolution - but perhaps this can be fixed at the editorial stage. 

I am satisfied that my other comments have been addressed satisfactorily. 

Best wishes, Dr Marie Murphy

[LINK]

---

## [Editor Report · Decision Letter 3]

2 Apr 2024

Dear Dr Barrientos-Gutierrez, 

On behalf of my colleagues and the Academic Editor, I am pleased to inform you that we have agreed to publish your manuscript "Predicted impact of banning non-essential, energy-dense food and beverages in schools in Mexico: a microsimulation study" (PMEDICINE-D-23-03317R3) in PLOS Medicine.

We also have one editorial request, which I will outline below.

PRESS

Sincerely, 

Katrien G. Janin, PhD 

Senior Editor 

PLOS Medicine

editorial request:

Apologies for having not picked up on this earlier, but I do find your authors' summary rather long and too result detailed. (Keep in mind that the Authors' Summary is intended for research to make findings accessible to a wide audience that includes non-scientists). I like to suggest to following summary to you. Please have a look and see if you feel that is an accurate non-technical reflection of your study. Please amend as needed.

"

Why was this study done? 

• School-based interventions have been recognized as effective means to improve nutritional knowledge and prevent obesity-related diseases. 

• In December 2023, the Chamber of Representatives of Mexico approved an amendment that strengthens and updates the General Education Law and nutritional guidelines to ban the sales and advertising of non-essential energy-dense food and beverages (NEDFBs) in schools. 

What did the researchers do and find? 

• The predicted impact of the intervention was modelled under different scenarios.

• We used age-specific equations to predict baseline fat-free mass and fat mass and total energy intake per day. We used microsimulation modelling to predict body weight and obesity prevalence of children and adolescents one year after implementing the intervention in Mexican schools. 

• Our modelling study suggests that an important impact on obesity prevalence can be expected if the law is implemented and enforced as intended. 

What do these findings mean? 

• If successful, this law could serve as an example beyond Mexico on how to achieve changes in obesity in children.

• The main limitation of our modelling study is that the study relies on the compliance of the schools. We also did not consider historical trends on obesity or NEDFBs consumed in schools during one year of simulation, and we considered only the ban impact inside schools, excluding effects near and outside schools. "